# Neuroendocrine Effects on the Risk of Metabolic Syndrome in Children

**DOI:** 10.3390/metabo13070810

**Published:** 2023-06-29

**Authors:** Giovanna Scorrano, Saverio La Bella, Sara Matricardi, Francesco Chiarelli, Cosimo Giannini

**Affiliations:** Department of Pediatrics, University of Chieti-Pescara, Via Dei Vestini, 66100 Chieti, Italy; giovanna.scorrano@studenti.unich.it (G.S.); saverio.labella@studenti.unich.it (S.L.B.); sara.matricardi@unich.it (S.M.); chiarelli@unich.it (F.C.)

**Keywords:** metabolic syndrome, obesity, neuroendocrine systems, stress response, genetic polymorphisms

## Abstract

The endocrine and nervous systems reciprocally interact to manage physiological individual functions and homeostasis. The nervous system modulates hormone release through the hypothalamus, the main cerebrally specialized structure of the neuroendocrine system. The hypothalamus is involved in various metabolic processes, administering hormone and neuropeptide release at different levels. This complex activity is affected by the neurons of various cerebral areas, environmental factors, peripheral organs, and mediators through feedback mechanisms. Therefore, neuroendocrine pathways play a key role in metabolic homeostasis control, and their abnormalities are associated with the development of metabolic syndrome (MetS) in children. The impaired functioning of the genes, hormones, and neuropeptides of various neuroendocrine pathways involved in several metabolic processes is related to an increased risk of dyslipidaemia, visceral obesity, insulin resistance, type 2 diabetes mellitus, and hypertension. This review examines the neuroendocrine effects on the risk of MetS in children, identifying and underlying several conditions associated with neuroendocrine pathway disruption. Neuroendocrine systems should be considered in the complex pathophysiology of MetS, and, when genetic or epigenetic mutations in “hot” pathways occur, they could be studied for new potential target therapies in severe and drug-resistant paediatric forms of MetS.

## 1. Introduction

Metabolic Syndrome (MetS) is a complex multifactorial disease with a not entirely recognized definition in childhood [1]. The International Diabetes Federation (IDF) published a consensus definition in 2006, according to which MetS is diagnosed in children between 10 and 15 years if at least three of the following risk factors occur: waist circumference > 90th percentile, systolic blood pressure > 130 mmHg or diastolic blood pressure > 85 mmHg, triglycerides > 150 mg/dL, and high-density lipoprotein (HDL) cholesterol < 40 mg/dL [2]. In adolescents older than 15 years, the adult criteria for MetS are used, while in children younger than 10 years, the MetS diagnosis is not validated [1]. In the paediatric population, excess fat storage is considered one of the most important risk factors for the development of both metabolic and cardiovascular diseases, including high blood pressure, atherosclerosis, dyslipidemia, and type 2 diabetes mellitus (T2D) [3]. Specifically, pediatric obesity is caused by both environmental factors and genetic susceptibility. In 2020, 39 million children under the age of five were overweight or obese worldwide, with some variations across countries, and up to 6.3% of children are currently affected by severe forms of obesity [4,5,6]. The most frequent form of pediatric obesity certainly results from a link between a susceptible genetic background and acquired environmental risk factors, which play a critical role in the pathophysiology [5,7,8]. Sedentary lifestyles represent an important acquired risk factor for overweight, obesity and MetS in children. Indeed, high-calorie diets, low sleep quality, a sedentary lifestyle, and a low socio-economic status are common conditions in obese children [5,7,8]. Notably, perinatal risk factors, such as a lack of breastfeeding and high birth weight, may also contribute to excessive fat storage accumulation [8,9,10]. Nevertheless, a considerable number of patients have a genetic form of obesity due to the abnormal functioning of different single genes and peptide hormones involved in the modulation of both hunger and satiety [5,7]. Some of these form part of the neuroendocrine system, including neuropeptides and peptide hormones that interact with various types of neurons and neuroendocrine signaling pathways [11]. These molecules are encoded by different genes, and their mutations can be responsible for genetically dependent excessive fat storage, according to neuroendocrine system abnormalities [5,11]. Furthermore, in several studies, some conditions were identified and related to disruption of the neuroendocrine system, with the subsequent risk of developing metabolic and cardiovascular disorders [11]. These include circumstances where the “stress system” is chronically and/or repeatedly activated, such as low birth weight, accidents, surgery, trauma exposure, physical or sexual abuse, behaviour disorders, anxiety, and depression [12]. Indeed, stress states are associated with the hyperactivation and disruption of neuroendocrine pathways involved in the stress response and in metabolic homeostasis control [12]. Even genetic polymorphisms and epigenetic modification, such as the DNA methylation of genes and neuropeptides involved in the stress response, are related to an increased risk of MetS development [13]. These mechanisms are able to directly affect the neuroendocrine pathways of the stress system and are associated with long-term biological consequences such as dyslipidaemia, hypertension, insulin resistance, and glucose intolerance [11]. Nevertheless, the metabolic risk in stressed paediatric patients is strictly related to personal history, environmental and developmental influences, the kinds of responses to stressors, and genetic background [13]. This review aims to describe the main neuroendocrine systems involved in the development of MetS in children, analysing the pathophysiology and the metabolic consequences of their disruption.

## 2. Neuroendocrine Pathways of the Stress Response and the Risk of Metabolic Syndrome

“Stress” is defined as a threatening condition where individual homeostasis is potentially endangered. In these circumstances, the organism activates neurobiological processes, known as “the stress response”, to face the threat and maintain homeostasis [14]. The stress system is mostly represented by the hypothalamic-pituitary-adrenal axis (HPA axis) and the arousal/sympathetic system at the central level, while the periphery effectors are constituted by glucocorticoids and catecholamines [15]. Thus, when activated, these systems modulate the endocrine, metabolic, immune, neuropsychiatric, and cardiovascular functions of many organs and tissues (Figure 1). An appropriate response to stressors and/or a limited exposure to stress induces the physiological processes that allow humans to face stressful conditions successfully [16]. Contrary, aberrant responses and long-term or very intense stress exposure are related to neuroendocrine responses that lead to crucial metabolic consequences [12]. Specifically, several animal models and clinical studies, have shown that inappropriate stress responses and chronic or intense exposure to stress lead to hyperactivity and consequent disruption of the stress system. These effects result in an increased risk of developing hypertension, insulin resistance, elevated blood glucose levels with decreased glucose uptake, visceral obesity and dyslipidaemia, T2D, an increased heart rate, higher blood pressure, and MetS [17]. Intense or chronic stress states include low birth weight, accidents, surgery, trauma exposure, physical or sexual abuse, behaviour disorders, anxiety, and depression. Interestingly, the crucial metabolic consequences of stress system dysregulation could be irreversible if they occur when brain plasticity is massive and growth is unrestrained, such as during the fetal period, childhood, and adolescence [12]. Indeed, in these crucial periods, HPA axis and autonomic nervous system disruption are characterized by delayed myelination, increased neuronal loss, and altered synaptogenesis in the central nervous system, with neuropsychiatric and endocrine consequences that persist even in adulthood [18]. 

### 2.1. Hypothalamic-Pituitary-Adrenal Axis 

The HPA axis plays a crucial role in stress response. During the stress response, the corticotrophin-releasing hormone (CRH), is secreted by the parvocellular neurons of the hypothalamus and stimulated by the neurons of the amygdala, hippocampus, mesocorticolymbic system, serotonin and acetylcholine systems, noradrenergic neurons of the sympathetic system, leptin, neuropeptide Y (NPY), arginine vasopressin (AVP) neurons in the hypothalamus, and inflammatory cytokines [12]. Recently, the pituitary adenylate cyclase-activating peptide (PACAP) was associated with CRH release after binding with its receptor, and expressed in the paraventricular nucleus of the hypothalamus [13]. CRH regulates the pituitary–adrenal axis and leads to secretion of the pituitary adrenocorticotropin hormone (ACTH) with the synergic role of AVP neurons of the paraventricular nuclei (PVN), that are slightly permissive for ACTH secretion. In physiological circumstances, CRH and AVP present a pulsatile circadian release, which is higher in the early morning [19,20]. On the contrary, during stress, an increased secretion of CRH and AVP occurs and is mediated by inflammatory cytokines, vasoconstrictor factors, and lipidic molecules that stimulate HPA axis activity. In the periphery, the main effectors of the HPA axis are glucocorticoids, secreted by the zona fasciculata of the adrenal cortex after the ACTH stimulus. During stress, glucocorticoid blood concentrations increase by up to ten times the baseline values. Indeed, studies on paediatric patients who were chronically stressed have shown higher plasma cortisol concentrations and increased sensitivity to glucocorticoids in the evening compared to healthy controls [21]. The primary mechanism of action of glucocorticoids is the regulation of gene transcription, which controls development, metabolism, and immune response when they bind to their receptor. These hormones increase cardiac output, catecholamine sensitivity, hepatic and renal gluconeogenesis, glycogen synthesis; decrease glucose uptake; and lead to increased central adipose tissue [22]. Physiologically, glucocorticoids carry out negative feedback on the HPA axis, leading to limited tissue exposure to these hormones without relevant effects. On the contrary, in conditions of chronic or intense acute stress, hyperactivation of the HPA axis occurs, with the disrupted glucocorticoid secretion that is associated with the long-term accumulation of fat in visceral adipose tissue, loss of muscle, arterial hypertension, osteoporosis, insulin resistance, dyslipidemia, and the potential development of MetS [15]. Indeed, repeated HPA axis activation causes the glucocorticoid receptor complex to translocate, which is followed by glucose-induced gene transcription. It leads, in turn, to increased expression of FK506 binding protein 51 (FKBP5) with a subsequent decrease in FKBP5-glucocorticoid receptor complex sensitivity. Indeed, FKBP5 is a co-chaperone of the glucocorticoid receptor, and when bound to the receptor complex, it decreases the affinity of glucocorticoids for their receptors, causing an ultra-short feedback loop [23]. The overexpression of FKBP5, after HPA axis hyperactivation, reduces the physiological glucocorticoid negative feedback to the HPA axis, enhancing hypercortisolism and its related metabolic consequences [13]. Previous studies indicated that patients with diabetes with poor blood glucose control, emotional stress, inflammation, or comorbidities presented HPA axis hyperactivation and hypercortisolism. Chronic activation of the HPA axis was also described in patients with type 1 diabetes with poor blood glucose control and in patients with T2D and diabetic neuropathy [24,25]. Therefore, there are several conditions associated with increased HPA axis activity that are potentially related to the risk of MetS. These include depression, eating disorders, obsessive-compulsive and panic disorders, chronic active alcoholism, attachment disorder in infancy, hyperthyroidism, and Cushing’s syndrome [12]. However, epigenetic and genetic factors, personal history, age, ethnicity, developmental and environmental influences, psychopathology, and the types of stressors could influence the HPA axis response with metabolic effects, and should be considered consequently [15].

### 2.2. Arousal and Sympathetic Nervous Systems 

Several studies have demonstrated the key role of the autonomic nervous system in the stress response. Specifically, during the stress response, noradrenergic neurons of the medulla and pons secrete norepinephrine at the locus coeruleus level, where the hormone is then transported to peripheral targets. The pons is the portion of the brainstem between the midbrain (rostral) and the medulla oblongata (caudal); the medulla is the lowest portion of the brain. Various structures within the pons and the medulla are responsible for regulating the autonomic nervous system as well as maintaining several vital body functions, including as heart function, and breathing. Noradrenergic neurons receive stimulatory innervation from neurons of the amygdala, hippocampus, mesocorticolymbic system, serotonin and acetylcholine systems, and CRH neurons, and they are also stimulated by leptin, inflammatory cytokines, and glucocorticoids [12]. All these structures are involved in the regulation of the autonomic nervous system via a complex and not completely understood signaling network [26]. The PACAP’s binding with its receptor in the hypothalamus also leads to CRH and norepinephrine release, sustaining the stress response [13]. At the peripheral level, the effectors of the sympathetic system are, therefore, norepinephrine and epinephrine. They reach the target organs (stomach, liver, pancreas, intestines, and adrenal gland), where they act through β-adrenoceptors, are involved in metabolic effects, and α-adrenoceptors, and mainly expressed on smooth muscle vessels [11]. Specifically, catecholamines stimulate hepatic glycogenolysis and gluconeogenesis and lead to lipolysis and fatty acid release from adipose tissue. Furthermore, the adrenal medulla secretes norepinephrine and epinephrine in response to stressors that lead to an accelerated heart rate, increased blood pressure and coronary flow, bronchial dilatation, inhibition of intestinal functions, and the stimulation of an inflammatory response [11]. Indeed, both the HPA axis and noradrenergic neurons at the locus coeruleus were shown to affect the release of proinflammatory cytokines with a sustained, latent, low grade of inflammation that sustains metabolic disruptions during stress states. The synergic interaction between CRH and noradrenergic neurons during the central stress response is sustained through α1-adrenoceptors and CRH type 1 receptors [12,27]. During the stress response, the sympathetic system is therefore hyperactive, and high levels of catecholamines are detected in blood and urine [17]. Interestingly, the hyperactivity of the sympathetic system occurring in stress states is related to metabolic effects, such as decreased glucose uptake, increased risk of insulin resistance, and visceral obesity with a β-adrenoceptor sensitization [28]. Moreover, catecholamines may contribute to insulin resistance by stimulating the overexpression of glucose transporters at the renal level, resulting in increased insulin secretion [11]. On the other hand, increased blood pressure and cardiac output are directly related to a high level of norepinephrine and a higher risk of developing cardiovascular diseases [29]. In several studies, it was observed that chronically stressed paediatric patients present increased levels of plasma and urine catecholamines compared to healthy controls [30]. The relevant metabolic consequences of the sympathetic system hyperactivation are therefore associated with an increased risk of hypertension, insulin resistance, T2D, visceral obesity, and MetS [11]. 

### 2.3. Genetic Variants and Epigenetic Modifications Involved in the Stress Response

Genetic and epigenetic backgrounds have a remarkable relevance in the stress response, with individual and variable consequences. Indeed, numerous genetic polymorphisms and epigenetic modifications are documented and considered responsible for individual genetic vulnerability [13,31]. This suggests that genetics, developmental influences, and environment modulate the stress response with different endocrine and metabolic involvements [32,33]. It was observed that genetic and epigenetic modifications affect the genes of the sympathetic nervous system, the HPA axis, and their mediators [11]. Specifically, the catechol-o-methyltransferase (COMT) gene regulates the expression and function of norepinephrine that represents the main hormone of the sympathetic nervous system. The COMT variant Val158Met was related to a phenotype with higher blood pressure and heart rate and visceral obesity [11]. Moreover, the gene haplo-insufficiency was associated with increased glycolysis and higher plasma cholesterol and triglyceride concentrations [34]. The β_1-2-3_-adrenoceptors (ADRB1, ADRB2 and ADRB3) genes encode the β-adrenoceptors, and their variants and/or epigenetic modifications are also associated with a high risk of MetS, leading to a genetic vulnerability during the stress response. Interestingly, ADRB1 polymorphisms were related to obesity, heart rate impairment, increased blood pressure, and heart failure [35]. On the other hand, ADRB2 variants were linked to insulin resistance, hyperleptinemia, obesity, and T2D, while ADRB3 variants have been associated with dyslipidaemia, obesity, and MetS [36]. Moreover, the solute carrier family 6-member 2 (SLC6A2) gene encodes the norepinephrine transporter, and its variants usually lead to decreased norepinephrine reuptake. Of note, associated symptoms include headache, palpitation, and tachycardia, with an increased cardiovascular risk. SLC6A2 activity is also related to metabolic effects, such as obesity and weight control, even though it has not yet been extensively studied [11]. Genetic polymorphisms in genes encoding HPA axis components are also reported in the literature. Specifically, the CRH receptors-1 and -2 (CRHR-1-2) genes are mainly expressed in the hippocampus, liver, and adipose tissue, and their variants were associated with depression, stress vulnerability, and cardiovascular impairment [11,37]. Genetic variants of genes involved in cortisol biosynthesis and catabolism were also described and related to MetS [11]. Interestingly, gene methylation was reported as the main epigenetic mechanism involved in the risk of MetS during the stress response [13]. Notably, FKBP5 plays a key role in the stress response, and its methylation was studied in depth [13]. FKBP5 is expressed in the brain, lymphocytes, muscle, and adipose tissue and modulates the nuclear transcription glucocorticoid-mediated, after translocation of the glucocorticoid receptor complex in the nucleus [38]. FKBP5 is related to decreased glucose uptake in omental tissue, insulin resistance, decreased levels of high-density lipoproteins (HDL), and stimulation of the inflammatory response [13,39]. It was observed that chronic hypercortisolism induced methylation of CpG sites on intron 2 of FKBP5, and patients with T2D, obesity, and insulin resistance show the same methylation pathway [13]. Furthermore, the nuclear receptor subfamily 3, group C, member 1 (NR3C1) gene encodes the glucocorticoid receptor and acts both as a transcription factor that binds glucocorticoid response mediators in the promoters of glucocorticoid target genes and as a regulator of other transcriptional factors. It was found that methylation mechanisms in this gene are associated with the development of MetS in stressed paediatric patients [11]. Prospective studies have shown different methylation profiles of interested genes in patients with insulin resistance and variable insulin and glucose blood concentrations [11,13]. Even though genetic and epigenetic modifications are influenced by several factors, such as individual development, environment, and social factors, they could be related to the risk of MetS, especially when they involve target genes.

## 3. The Regulation of Satiety and Hunger in Central Neuroendocrine Pathways

The central neuroendocrine system plays a key role in the balance of satiety and hunger, and in both decreasing and increasing food intake and fat storage deposition. The leptin–melanocortin pathway is the most important pathway promoting satiety and fat accumulation, stimulating the production and cleavage of pro-opiomelanocortin (POMC) [5,40]. Meanwhile, the NPY/agouti-related protein (AgRP) neuroendocrine system is critical in encouraging food intake and hunger [5,7,11]. The hypothalamus is a central brain region for the correct regulation of satiety and hunger, especially the arcuate nucleus, with both POMC and NPY–AgRP neurons, and the paraventricular nucleus, with melanocortin neurons expressing melanocortin receptors (MCRs) such as the melanocortin 4 receptor (MC4R) [5,7]. Below, a detailed description of these important neuroendocrine pathways and their relation to metabolic consequences is presented.

### 3.1. Leptin and LEPR, a Key Neuroendocrine Tandem with Metabolic Implications

Leptin is a protein encoded by the gene *Lep* (7q31.3), which is regarded as a crucial molecule in the stimulation of satiety and the promotion of energy expenditure [5,7,41,42]. Leptin is produced primarily by white adipose cells in proportion to fat mass, but it is also secreted by neurons in the hypothalamus and pituitary [11,42,43]. Low leptin levels correlate with a poor energy state for the organism, eliciting responses to protect and enhance the energy reserve, including altered behavioral and metabolic adaptation mechanisms [44]. Instead, obese children and adults have higher leptin levels than healthy individuals [4,42,45]. Leptin is an integral part of the neuroendocrine system due to its ability to cross the blood–brain barrier and interact with its receptors LEPRs, which are highly expressed by GABAergic neurons in the arcuate nucleus of the hypothalamus [4,7,11,42,46]. The binding of leptin and LEPRs in the arcuate nucleus stimulates the expression of POMC by a large population of POMC neurons, resulting in its cleavage into melanocortins, reducing hunger (Figure 2) [7,47,48]. Notably, the SH2B adaptor 1 (SH2B1) protein was identified as a key adapter in the leptin-induced signal transduction of POMC, with clinical implications [49,50,51]. Furthermore, the binding of LEPR and leptin inhibits NPY–AgRP neurons via a distinct neuroendocrine pathway, reducing food intake and fat accumulation [4,7,11,42,46]. Due to its ability to stimulate the sympathetic nervous system and promote adrenoceptor activity, chronic leptin infusion also causes systemic inflammation and increases blood pressure [11,52]. This reveals the multiple interconnections between leptin and the cardiovascular system and explains why leptin deficiency has numerous metabolic implications in children and adults [11,53]. Insulin resistance, dyslipidemia, and visceral obesity were observed in obese patients with leptin impairment [5,7,11]. Indeed, some forms of monogenic obesity brought on by loss-of-function *LEP* mutations are associated with severe childhood obesity, hyperphagia, and, frequently, T2DM [41,54,55]. In recessive forms, it is also possible to observe hypothyroidism, central hypogonadism, and delayed puberty [41,53,55]. LEPRs also play a crucial role in the satiety neuroendocrine system [5,7,11,40]. These neuronal receptors are encoded by the gene *LEPR* (1p31.3) and widely expressed in the arcuate nucleus of the hypothalamus with different implications, as discussed above, including the promotion of POMC expression and a reduction in NPY–AgRP activity [5,11]. However, LEPRs were also detected in leptinergic neurons of the rostroventral lateral medulla, where they regulate blood pressure and the sympathetic system via interactions with the kidneys [11,56]. Extremely rare, affecting a few more than 80 patients worldwide, pathogenic *LEPR* mutations are linked to severe childhood obesity, hyperphagia, hyperinsulinemia, hyperlipidemia, and, less frequently, multiple pituitary hormone deficiencies [5,40,57,58].

### 3.2. The Key Role of POMC, Melanocortins, and Related Receptors in the Development of MetS

POMC neurons are primarily found in the arcuate nucleus of the hypothalamus, where they are key neuroendocrine actors in the leptin-induced modulation of appetite and food intake [7,48]. POMC is encoded by the gene POMC (2p23.3) in response to leptin and LEPR binding stimulation and represents the precursor hormone peptide of melanocortins [5,7,11]. POMC is metabolized into multiple melanocortins via the cleavage of proprotein convertase subtilisin/kexin type 1 (PCSK1) and the cooperation of carboxypeptidase E (CPE) and steroid receptor coactivator 1 (SRC-1) (Figure 3) [7,41,48,59,60]. Melanocortins are a large family of tiny neuropeptides derived from POMC, which includes the ACTH and several types of smaller melanocyte-stimulating hormones (MSH), such as α-MSH, β-MSH, and two γ-MSH isoforms [11,61]. ACTH is essential for the regulation of cortisol, which is produced by the adrenal gland. Other neuroendocrine molecules produced by ACTH cleavage include endorphins such as proenkephalin-A and -B, met-enkephalin, β-lipotropin and γ-lipotropin, which are essential for lipolysis, lipid transport, and steroidogenesis [11]. MSH interacts with its transmembrane G-coupled melanocortin receptors (MCRs) in the hypothalamus, and, among these, MC4R and MC3R are the most important. Particularly noteworthy is the MC4R’s ability to bind not only melanocortins but also AgRP [5,11,61]. MC4R and MC3R are widely distributed: MC4Rs are expressed in the paraventricular nucleus and amygdala and aim to regulate food intake, whereas MC4Rs elsewhere regulate energy consumption [11,62]. MC4R may play a critical role in the pathophysiology of MetS. Indeed, the interaction between melanocortins and MC4R results in the expression of various genes with the goal of regulating body weight and energy expenditure, with the aid of the transcription factor single-minded homologue 1 (SIM1) [5,11,63,64,65,66]. In addition, the melanocortin–MC4R pathway is also essential for regulating cholesterol metabolism, particularly the HDL/LDL ratio [11,67]. Pathogenic mutations affecting the gene *MC4R* (18q21.32) can be considered as the greatest risk factors for the co-occurrence of obesity and T2D, with MC4R deficiency being the most common genetic cause of obesity, affecting nearly five percent of all obese children [11]. Indeed, several autosomal-dominant (AD) and autosomal-recessive (AR) loss-of-function mutations of *MC4R* cause hyperphagia, severe pediatric obesity, and hyperinsulinemia due to peripheral receptorial impairment [64,66,68]. Furthermore, gain-of-function mutations of *MC4R* were found to improve body mass index and to decrease appetite, cardiovascular complications, and other metabolic features [5,11,69]. MC3R also has a role in the relationship between neuroendocrine systems and MetS. Unlike other melanocortin receptors, MC3R is predominantly found in the limbic and hypothalamic regions of the brain, including the ventromedial hypothalamus, with the aim of regulating metabolic homeostasis and food intake [11]. Pathogenic mutations of the gene *MC3R* (20q13.2) are also responsible for the accumulation of fat mass, diet-induced obesity, cycling hyperinsulinemia, and glucose intolerance [11]. Conversely, pathogenic mutations affecting POMC can result in genetically related severe obesity and MetS with pediatric onset [5,70]. Hence, the phenotype of POMC-associated monogenic obesity is typically characterized by pale skin and red hair, hyperphagia, severe obesity, adrenal insufficiency, and occasionally cholestasis [70,71]. Interestingly, maternal diet during fetal development has a significant effect on the DNA methylation of POMC, underscoring the critical importance of environmental factors in regulating the leptin-melanocortin pathway and the neuroendocrine systems [5,72]. However, the mutations in enzymes and adaptor proteins implicated in these neuroendocrine pathways may result in abnormal phenotypes with metabolic implications. In fact, loss-of-function mutations of *PCSK1* have been related to severe obesity, malabsorptive diarrhea, adrenal insufficiency, hypogonadism, hypothyroidism, and postprandial hypoglycemia [5,7,73]. Instead, clinical pictures due to loss-of-function mutations of the gene *SH2B1* (16p11.2) are characterized by childhood obesity, insulin resistance, and hyperphagia [49,50]. In addition, loss-of-function mutations of the gene *CPE* (4q32.3) can inhibit the cleavage of POMC into melanocortins, resulting in a triad composed of severe obesity, developmental delay, and hypogonadotropic hypogonadism [59,74]. The early onset of severe obesity, hyperphagia, persistent diarrhea, and metabolic abnormalities may be linked to mutations in the gene *SRC-1*, that inhibit the leptin-induced synthesis of POMC [60,75]. Lastly, haploinsufficiency of the gene *SIM1* (6q16.3) can result in a severe form of genetic obesity in children [65]. Intriguingly, *SIM1* plays a crucial role in the development of the *Drosophila* brain, and its impairment was linked to obesity and high leptin levels in mice, although the underlying mechanisms are not entirely clear, providing further evidence of the connections between neuroendocrine signaling pathways and MetS [65,76,77]. 

### 3.3. Ghrelin, the NPY–AgRP Pathway, and Metabolic Implications

Even though the most primary feature of ghrelin is to stimulate the secretion of growth hormone (GH), it is also considered the most important hunger-stimulating hormone, counteracting leptin’s effects. Ghrelin is encoded by the gene *GHRL* (3p25.3) and produced by neuroendocrine cells in the gastric mucosa and pancreas, as well as by neurons in the hypothalamus and anterior pituitary. Its effects are mediated by NPY–AgRP neurons in the arcuate nucleus, whose aim is to antagonize MC4Rs. Interestingly, low levels of ghrelin are correlated with MetS, while high concentrations seem to be protective against cardiovascular risk [11,78]. The ghrelin receptor (GHS-R), which is expressed by cells of the nervous and gastrointestinal tract, is also involved in the neuroendocrine system, with effects on lipid and glucose metabolism. The primary effect of ghrelin is the stimulation of orexigenic NPY–AgRP neurons in the arcuate nucleus, promoting their neuroendocrine pathway and whose purpose is to promote appetite and fat mass deposition [5,11]. NPY–AgRP neurons are a large population of hypothalamic neurons that secrete AgRP and NPY, which inhibit the binding of melanocortins to MC4R [11]. NPY–AgRP neurons are tightly regulated, as their expression is inhibited by leptin via the expression of LEPR and stimulated by ghrelin, brain-derived neurotrophic factor (BDNF), and its receptor encoded by the neurotrophic tyrosine kinase receptor type 2 gene (NTRK2) [5]. The NPY–AgRP neuroendocrine system is regarded as an important neurophysiological link between sympathetic stress reactivity, the glucocorticoid system, and the melanocortin system [11]. In fact, NPY–AgRP neurons release γ-aminobutyric acid (GABA) to inhibit POMC neurons. AgRP is produced by a large group of neurons in the arcuate nucleus and paraventricular nucleus, which express both ghrelin and leptin receptors [11]. AgRP enhances food intake and appetite in response to pro-inflammatory cytokines by inhibiting the interaction between MSH and MC4R [5,11]. NPY is widely present throughout the cerebrum, although the hypothalamus and limbic system are its primary sites of activity [11,79]. Indeed, NPY is primarily produced in the arcuate nucleus, with its orexigenic effects manifesting in the paraventricular nucleus, thereby inducing hunger, lowering energy expenditure, and contributing to obesity [11]. Some of the NPY receptors mediate neuroendocrine effects and, among these, NPY1R expression in fat cells is associated with the development of MetS by promoting insulin resistance and fat mass accumulation, whereas some variants of NPY2R may influence cardiometabolic traits [11,79,80,81]. In addition, the gene *NPY2R* (4q31) plays a crucial role in an animal model of MetS, causing abdominal fat deposition and angiogenesis in adipose tissue in response to sympathetic nerve stimulation in response to social or environmental stress [11,82]. NPY’s metabolic effects include vasoconstriction and a reduction in blood pressure via direct stimulation of potassium channels and intracellular calcium, as well as a promotion of fat deposition [11,82]. In addition, the release of NPY by paraventricular neurons may contribute to the complex pathophysiology underlying the onset of T2D [11]. Notably, a specific leucine7 to proline7 (Leu7Pro7) polymorphism and the single nucleotide polymorphism rs16147 of the gene *NPY* (7p15.1) were linked to elevated cholesterol and abdominal fat deposition, confirming the central role of this neuroendocrine system in the MetS [11,83,84]. 

Overall, the available data clearly show that significant progress has been made in recent years, including a better understanding of the molecular mechanisms underlying autonomic stress responses and the neuroendocrine signaling pathways linked to MetS. However, more relevant studies, particularly concerning childhood, are needed to better describe the stress pathway associated with the development of MetS in children, to provide therapies that are more precise for patients with the above-discussed genetic dysfunctions, and to better understand the pathophysiology underlying genetic background and molecular pathways. However, a number of issues must be clarified, primarily pertaining to the complex regulatory strategies of these pathways by the nervous and endocrine systems. A novel promising treatment target appears, for example, in the protein neudesin, a recently discussed neuropeptide that, in obese and overweight children, appears to be significantly correlated with blood glucose [85]. However, a more detailed characterization of the main pathogenic variants of the critical genes involved in MetS in children should be performed. Finally, an additional and relevant limitation of our review is the lack of data characterizing the main differences between children and adults showing the peculiarities well-described in obesity childhood compared to adulthood [86]. Thus, studies characterizing this relevant topic, highlighted in this review, are needed to further pave the way for potentially novel strategic targets in MetS.

## 4. Conclusions

Several neuroendocrine pathways involved in metabolic processes play a key role in maintaining individual homeostasis, and their disruption is associated with cardiovascular and metabolic disorders. Specifically, chronic, or intense acute stress, impairs the neuroendocrine systems of the stress response, leading to their hyperactivation. At the periphery, catecholamine and glucocorticoid activity increase, with an elevated risk of dyslipidaemia, hypertension, visceral obesity, insulin resistance, and glucose intolerance. Genetic and epigenetic modifications of genes that encode the neuroendocrine pathways of the stress response also influence the individual’s genetic vulnerability and the cardiometabolic risk related to the stress states. In addition, central neuroendocrine pathways play their own role in promoting satiety or hunger states. The leptin–melanocortin pathway and the ghrelin-NPY-AgRP neuroendocrine system have different roles but are both critical for a correct metabolic balance. Neuroendocrine systems should be considered in the complex pathophysiology of MetS, and, when genetic or epigenetic mutations in “hot” pathways occur, they could be studied for new potential target therapies in severe and drug-resistant paediatric forms of MetS. However, the underlying mechanisms and genetic backgrounds behind paediatric forms of MetS and related to the impaired functioning of neuroendocrine systems should be further studied and better understood in order to allow for a full comprehension of these critical pathways in human pathophysiology.

## Figures and Tables

**Figure 1 metabolites-13-00810-f001:**
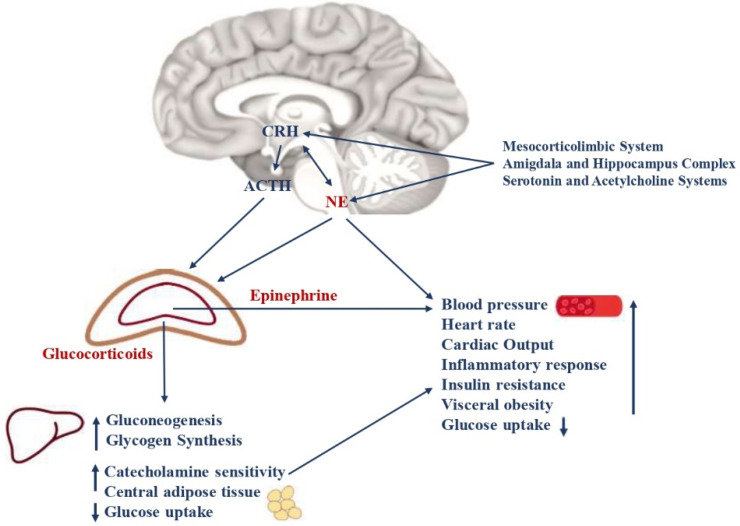
Neuroendocrine pathways of the stress response. The stress system consists of the hypothalamic-pituitary-adrenal axis (HPA axis) and the arousal/sympathetic system at the central level, while the peripheral effectors are represented by glucocorticoids and catecholamines. The two systems interact with each other at different levels. CRH and NE are the main central mediators and receive stimulatory afferents from the amygdala/hippocampus complex, the mesocorticolimbic system, serotonin, and acetylcholine systems. CRH is secreted by the parvocellular neurons of the hypothalamus, while NE is mainly synthesized in the medulla and pons at the locus coeruleus level. They, in turn, stimulate the anterior pituitary gland to secrete ACTH and the adrenal medulla to secrete epinephrine and norepinephrine, respectively. The ACTH release leads to glucocorticoid synthesis by the cortex of the adrenal gland. Catecholamines and glucocorticoids then act in synergy on target organs and tissues. The main metabolic consequences are gluconeogenesis, glycogen synthesis, decreased glucose uptake, insulin resistance, visceral obesity, a low grade of chronic inflammation, and a higher risk of glucose intolerance, type 2 diabetes, and MetS. Moreover, catecholamines lead to increased cardiac output, heart rate, and blood pressure, with a higher risk of hypertension and cardiovascular diseases. CRH = corticotrophin-releasing hormone; NE = norepinephrine; ACTH = adrenocorticotropin hormone; MetS = metabolic Syndrome.

**Figure 2 metabolites-13-00810-f002:**
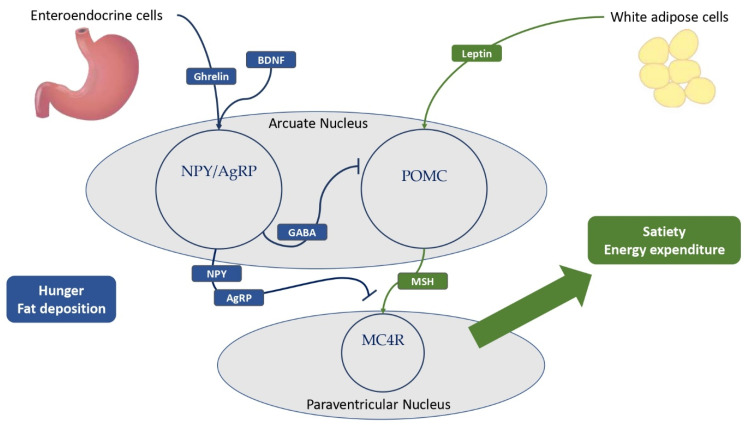
The main hypothalamic neuroendocrine pathways involved in hunger and satiety. Ghrelin is mostly produced by the neuroendocrine cells of the gastrointestinal tract. White adipose cells are the primary source of leptin production, and obese patients have higher levels of leptin compared to normal-weight individuals. Ghrelin and BDNF stimulate NPY–AgRP neurons to express GABA, thereby inhibiting POMC neurons in the arcuate nucleus of the hypothalamus. In addition, NPY and AgRP antagonize MSH-MC4R binding to a large family of neurons in the paraventricular nucleus of the hypothalamus, resulting in hunger and fat deposition. In addition, leptin stimulates the cleavage of POMC to melanocortins, promoting satiety and energy expenditure. BDNF = brain-derived neurotrophic factor; NPY = neuropeptide Y; AgRP = agouti-related protein; GABA = γ-aminobutyric acid; POMC = pro-opiomelanocortin; MSH = melanocyte-stimulating hormones; MC4R = melanocortin receptor type 4.

**Figure 3 metabolites-13-00810-f003:**
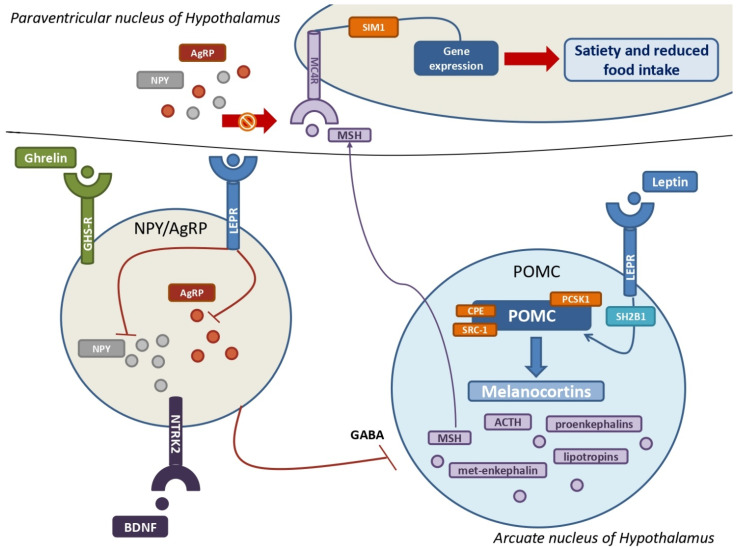
The leptin-melanocortin pathway and its interactions with the NPY–AgRP neuroendocrine system. A huge population of neurons in the arcuate nucleus produce pro-opiomelanocortin (POMC), which is then cleaved into melanocortins by PCSK1, CPE, and SRC-1 upon leptin and its receptor (LEPR) binding. SH2B1 is a crucial adaptor protein involved in the leptin-LEPR pathway. The various MSH isoforms bind MC4Rs, which are highly expressed by MC4 neurons in the paraventricular nucleus of the hypothalamus, resulting in multiple gene expressions that promote satiety and decreased food intake. The stimulation of NYP/AgRP neurons by ghrelin, BDNF, and their own receptors results in a massive production of both NPY and AgRP. Their main goal is to antagonize MSH from binding with MC4Rs to increase hunger and improve food intake. Leptin also inhibits NPY–AgRP neurons and their suppression of POMC neurons via GABAergic systems. POMC = pro-opiomelanocortin; PCSK1 = proprotein convertase subtilisin/kexin type 1; CPE = carboxypeptidase E; SRC-1= steroid receptor coactivator 1; LEPR = leptin receptor; SH2B1 = SH2B adapter protein 1; MSH = melanocyte-stimulating hormones; MC4R = melanocortin receptor type 4; NPY = neuropeptide Y; AgRP = agouti-related protein; GABA = γ-aminobutyric acid.

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
