# Peer review of "Neuroendocrine Effects on the Risk of Metabolic Syndrome in Children"

_metabolites, 2023, doi:10.3390/metabo13070810_

Round 1

Reviewer 1 Report

The manuscript by Giovanna Scorrano et al provides a comprehensive and well-written review of the neuroendocrine effects on the risk of metabolic syndrome in children.

Minor points:

There is no need for some words such as Medulla, Pons, Hypothalamus Amygdala, Hippocampus, Mesocorticolymbic System, Serotonin, Acetylcholine Systems to be capitalized in some parts of the manuscript?

Line 302 (fig. 2) – please change to (Figure 2).

Author Response

Thank you for your suggestion.

Fig.2 has been changed in Figure 2. In addition, through the manuscript as suggested, we have changed the capital letter for all these words. Finally, we have added a few sentences explaining the significance of these structures, as well as a new reference detailing the physiology of the central nervous system and autonomic nervous system (doi:10.1038/nrn2647), [26].

Reviewer 2 Report

 The review "Neuroendocrine effects on the risk of metabolic syndrome in children" presented by Giovanna Scorrano is a resume on possible metabolic risk factors that affect health of children at young age. The information is focused on the subject and gives a decent overview of the state-of-the-art. Though in parts the writing changes from easy reading to dense parts the comprehensive graphic illustrations helps the reader to resume the information and maintain the overview. All together the text is a nice resumed paper for the expert and the non-expert reader.

Overall, I like the presented work, but my main criticism is that there is little or no opinion of the authors on the state-of-the-art. I would like to have seen some indications on what is missing currently in the diagnosis, why is it so and what can be done to improve the situation. Without this type of comments, the review lacks an identity and won´t make a difference with respect to other review on this or similar subjects. 

Author Response

Thank you for your comment. We added some sentences at the end of the discussion to provide the authors' perspective and give this manuscript a proper identity.

Reviewer 3 Report

In this review article the auhtors examine the current literature on the effects of neuroendocrine pathway (mainly hypothalamic) disruption on the risk of MetS "in children", with a special focus on the biology of the stress response. The authors propose that neuroendocrine systems should be considered in the complex pathophysiology of MetS, and, that genetic or epigenetic mutations in "hot" pathways should be studied in search for new potential target therapies in severe and drug-resistant pediatric forms of MetS. The manuscript is well written and well organized and easy to follow. The topic is of great current interest. I have some observations for the authors:

1. Although the authors claim that the discussed pathways are relevant to pediatric patients, actually they are relevant to the metabolic syndrome at (almost) all ages. I acknowledge that some literature comes from studies in children but it is unclear how the whole review applies to children.

2. The review is a general good landscape of the topic and thus it did not go into detail due to the wide focus of the literature review. There are no specific examples of what new specific targets can be exploited for the treatment of the MetS in children.

3. To a certain degree, this review relies somewhat heavily on previously published reviews rather than original papers. What is the authors explanation for that.

4. The authors did not comment on the limitations of their literature review.

Author Response

  • Although the authors claim that the discussed pathways are relevant to pediatric patients, actually they are relevant to the metabolic syndrome at (almost) all ages. I acknowledge that some literature comes from studies in children but it is unclear how the whole review applies to children.

Thank you for your comment. This is a crucial aspect of the manuscript that we have attempted to achieve. We agree with the comment that the pathways are relevant in all ages not only in childhood but the effects largely described in the manuscript with also references of studies conducted in childhood and definition of MetS adopted only in children certainly have an higher impact in this crucial age. In fact, we need to acknowledge that pediatric age is peculiar due to the fact that during growth children physiologically change the body composition and activation. Thus, alteration of these pathways might play a different role in childhood compared to adulthood. In fact, childhood obesity is certainly related to obesity risk in adulthood and the development of complications such as T2D or other obese related complications, namely MetS is faster and more severe compared to adulthood as discussed in a previous review (Giannini C et all Hormone research in pediatrics Horm Res Paediatr. 2022;95(2):149-166). Finally, to focus the paper mainly on data in children, in the manuscript, we have provided specific data for the epidemiology of childhood obesity, discussed the different classification criteria for MetS in children, and outlined the genetic and environmental factors that can lead to pediatric MetS such as LEPR mutations, MC4R pathogenic heterozygous and homozygous variants, POMC-, SH2B1-, CPE, and other genes involved in neuroendocrine pathways and related to monogenic childhood obesity.

  • The review is a general good landscape of the topic and thus it did not go into detail due to the wide focus of the literature review. There are no specific examples of what new specific targets can be exploited for the treatment of the MetS in children.

Thank you for your comment. Although the scope of this review is not finalized to treatment strategies, we read with great interest a recent cross-sectional preliminary study conducted on the role of the neuropeptide Neudesin as potential target of novel drugs for MetS in children (10.3389/fendo.2022.881524); we added this point in lines 442-444. However further studies evaluating the potential target for treatment are needed to push further in the treatment options.

  • To a certain degree, this review relies somewhat heavily on previously published reviews rather than original papers. What is the authors explanation for that.

Thank you for your comment. We have attempted to provide a narrative overview of the relevant literature. We think that our review represents a comprehensive overview of all the available data. Thus, data including review and clinical studies have been evaluated with the aim not to replace already published review but just to summarize all available reports in a topic that is still too novel and that need to be stressed in clinical research and case report. If achieved these goals might allow to completely define the role of physiological anti-stress pathways that might open new therapeutic strategies in MetS in childhood and obesity. Nonetheless, a large number of original papers have been utilized in the study of these molecular pathways, primarily for the characterization of genetic mutations involving the genes. In addition, we included three original figures based on the major signaling pathways discussed in the text in order to enrich the reader's understanding and facilitate comprehension of the complex issues addressed in our review and other papers.

  • The authors did not comment on the limitations of their literature review.

Thank you for your comment. This was stated in the discussion's final sentence (lines 434-451).